# The Colorimetric Detection of the Hydroxyl Radical

**DOI:** 10.3390/ijms24044162

**Published:** 2023-02-19

**Authors:** Yandong Ran, Mohammed Moursy, Robert C. Hider, Agostino Cilibrizzi

**Affiliations:** 1Institute of Pharmaceutical Science, King’s College London, London SE1 9NH, UK; 2Centre for Therapeutic Innovation, University of Bath, Bath BA2 7AY, UK

**Keywords:** hydroxyl radical, iron complexes, fatty acids, aromatic hydroxylation, Fenton reaction, Fenton activity

## Abstract

An aromatic substrate for hydroxylation by hydroxyl radicals (^•^OH) was investigated. The probe, *N,N*’-(5-nitro-1,3-phenylene)-*bis*-glutaramide, and its hydroxylated product do not bind either iron(III) or iron(II), and so they do not interfere with the Fenton reaction. A spectrophotometric assay based on the hydroxylation of the substrate was developed. The synthesis and purification methods of this probe from previously published methodologies were improved upon, as well as the analytical procedure for monitoring the Fenton reaction through its use, enabling univocal and sensitive ^•^OH detection. The assay was utilised to demonstrate that the iron(III) complexes of long-chain fatty acids lack Fenton activity under biological conditions.

## 1. Introduction

Redox-active metals such as iron and copper are believed to play major roles in diseases such as inflammation and β-thalassaemia [1,2]. These metal ions are capable of generating the highly damaging hydroxyl radical (^•^OH) via the Fenton reaction (Equation (1)): Fe^2+^ + H_2_O_2_ → Fe^3+^ + ^−^OH + ^•^OH(1)

The ^•^OHs generated by the Fenton reaction react with most biomolecules, inducing hydroxylation, hydrogen and/or other substituent abstraction and replacement, or electron transfer reactions. Moreover, they may in turn lead to the formation of secondary radicals, which cause the chemical modification and subsequent damage of proteins, lipids, carbohydrates, and nucleotides [3]. Iron(III) in solution tends to aggregate as insoluble polynuclear complexes at physiological pH values; water-soluble ligands are therefore required to keep iron(III) in solution. Iron chelated to ligands such as citrate, amino acids, and organic phosphates may induce ^•^OH formation in vivo.

Because of the high reactivity of ^•^OH, its detection in biological systems is extremely difficult. A number of biochemical assays are available for ^•^OH detection. These methods include the production of formaldehyde from dimethylsulfoxide [4], the production of ethylene from methional [5], the oxidation of formate to carbon dioxide [6], the bleaching of *p*-nitrodimethylaniline [7], the formation of hydroxylated aromatic products [8,9,10,11], the use of the spin trap labels [12], and the application of fluorescent probes [13,14]. With the exception of aromatic hydroxylation, these methods are either non-specific for ^•^OH or dependent on elaborate instrumentation.

With regard to aromatic substrates, the ^•^OH reacts predominantly by addition to the ring and not by interaction with the substituents [15,16]. The position of attack on the ring may not always be univocal and is usually influenced by the specific molecular features [17,18]. In general, it depends on the electron-donating/withdrawing properties of other substituents present [19]. Theoretically, due to the electrophilic character of ^•^OH, an attack occurs at positions on the ring that are activated by electron-donating substituents, i.e., *ortho* and *para* positions. In contrast, in the case of nitrobenzene, the nitro group is instead a so-called ‘deactivating substituent’, which would favour ^•^OH electrophilic substitution in the *meta* position. Nonetheless, the possible formation of various hydroxylation products (including mono-hydroxylated and di-hydroxylated species) via nonspecific and nonstoichiometric reactions has also been reported for the reaction of ^•^OH with aromatic substrates [18,20]. Thus, there are a number of assays which take advantage of the ability of ^•^OH to attack aromatic rings, for example, salicylate hydroxylation (Equation (2)) [9], but these would require a strict control of experimental conditions to limit analytical bias and/or artefacts.

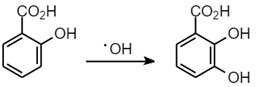
(2)

Although such assays are useful, they suffer from several disadvantages. For instance in the case of the salicylate hydroxylation assay, it can lead to the formation of catechol species, which, under incubation conditions, will bind iron(III) to form 2:1 complexes [20]. The binding of iron to the newly formed catechol may produce additional ^•^OH and will certainly perturb the reaction under observation. A further problem with salicylate is that it has a relatively high affinity for iron(III), and such complexation will inevitably perturb the analysis [21]. Clearly, the binding of iron by either the substrate or the hydroxylated products is undesirable, as both will influence the hydroxyl-radical-generating ability of a given system.

A strategy to avoid these difficulties was previously reported [22]; however, we found difficulty in following both the synthesis and the analytical procedure described in this communication. We now report a reliable method for the synthesis of the reagent *N,N*-(5 nitro-l,3-phenylene)-*bis*-glutaramide (NPBG) and the conditions for its use for ^•^OH detection.

## 2. Results

### 2.1. Design and Synthesis of the Probe

In order to overcome the problems associated with iron chelation by either the probe or its hydroxylated product, Singh and Hider [22] designed a novel substrate. This substrate (NPBG, Figure 1) is substituted in such a manner as to prevent the formation of an *ortho* dihydroxy function in the oxidised product (as in the case of salicylate hydroxylation). 1,3,5-trisubstitution avoids this possibility. The nitro group enhances the sensitivity of the UV absorbance spectral change, due to its strong electron withdrawing character and the capability of resonance stabilisation provided to the hydroxylated products. 

We experienced difficulty in achieving the previously reported yields [22], and so the synthesis (Figure 1) was modified as detailed in the Materials and Methods section. Briefly, the reaction time for the synthesis of 3,5-diaminonitrobenzene was extended to 30 min, and the isolation of the product was achieved by extraction into ethyl acetate, followed by column chromatography. This procedure ensured the removal of traces of the starting material from the product. Conversion of 3,5-diaminonitrobenzene into NPBG was achieved using the reported method [22].

### 2.2. Development of the Assay for the Detection of ^•^OH

Initially, we adopted the previously published conditions [22] but experienced a number of complications. The sequence of the addition of reagents to the borate buffer proved to be critical. It is essential to dissolve the iron salt in unbuffered water initially, followed by the ligand under investigation. Only then should the solution be adjusted to pH = 8.0, with the addition of borate buffer. Otherwise, the variable extent of iron hydroxide formation leads to a non-consistent spectrum. Indeed, even using the following sequence, we obtained reproducible data only after brief centrifugation of the samples: dissolution of chelator first, followed by the addition of the iron salt. These modifications led to a reliable assay which produced UV–visible spectra with a maximum of 435 nm (Figure 1A). There was a linear increase in absorbance (R^2^ = 0.99) over the period of 1 h (Figure 1B). A fixed timed assay was adopted for the study.

The hydroxylated products were resolved by semi-preparative HPLC, as reported in the original reference [22]. Further details and spectral data are reported in Appendix A. The predominant products formed upon hydroxylation are the *ortho-* and *para-*hydroxy nitro-compounds. The two isomers were characterised by measuring their molar absorption coefficient and λ_max_ values, *o-*hydroxy product, λ_max_ = 330 nm (ε = 2700), and the *p-*hydroxy product λ_max_ = 300 nm (ε = 8500). Only the mono-hydroxylated derivatives where detected, and they are formed at an approximate ration of 6:1.

### 2.3. Comparison of the ^•^OH Generating Ability of Various Complexing Agents

The following ligands were compared: EDTA (ethylenediaminetetraacetic acid), DTPA (diethylenetriamine pentaacetate), NTA (nitrilotriacetic acid), citrate, maltol, and deferiprone (Figure 2). The range of log affinity constants for iron(III) fell between 35.7 and 11.4, and for iron(II), it fell between 16.4 and 4.4 (Table 1). Under the conditions of the study, the rates of hydroxylation ranged from 13.59 to 0.37 mol h^−1^.

The iron(III) complex of EDTA, which is well established to generate Fenton chemistry, was observed to produce one of the highest rates, as did NTA. In contrast, citrate, which forms oligodentate iron(III) complexes [25], did not. In a similar fashion, deferiprone, a widely used clinical iron chelator [26], also failed to produce high rates of hydroxylation. It is well established that the 3:1 deferiprone–iron(II) complex is not Fenton active [27]. Maltol, which is a similar bidentate ligand to deferiprone [26], but with a weaker affinity for iron(III) (Table 1), was found to induce a low rate of hydroxylation, albeit higher than that of deferiprone. The gradual decrease in the rate of hydroxylation with NTA as the molar ratio is increased from 1:1 to 5:1 is consistent with the predominance of the [NTA·Fe] complex at low ligand:Fe ratios and the predominance of the [(NTA)_2_Fe] complex at high ligand:Fe ratios. The coordination sphere of iron in the [NTA·Fe] complex is more accessible to H_2_O_2_ (i.e., the 1:1 complex would have a vacant coordination site for attack) than that of the [(NTA)_2_Fe] complex. Thus, the trends in the changes observed in the rates of hydroxylation can be readily explained in terms of the established properties of the iron(III) chelators. 

The marked difference in the rate of hydroxylation induced by iron(II) and iron(III) in the presence of DTPA was surprising. DTPA probably binds iron(III) in a septadentate mode, preventing the access of H_2_O_2_, in contrast to the analogous EDTA, which binds iron in hexadentate mode, thereby leaving part of the iron coordination sphere exposed to attack by H_2_O_2_ (similarly to [NTA·Fe] complex) [24]. The enhanced kinetic lability of iron(II) accounts for the much higher Fenton activity of the iron(II) DTPA complex.

### 2.4. Comparison of the ^•^OH-Generating Ability of Various Fatty Acids

Fatty acids are established to enhance the absorption of iron(III) [28,29], and, as they are a major food source, it is an attractive concept to consider fatty acid iron(III) complexes as a vehicle for the oral administration of iron in various types of anaemia. The advantage of using iron(III) salts in the place of iron(II) salts is that, generally, preparations with the former are less Fenton active than those with the later and are therefore generally less toxic [30]. However, the bioavailability of iron when supplied in the iron(III) oxidation state is generally lower than when provided in the iron(II) state [31]. As there are reports of fatty acids facilitating the intestinal absorption of iron [32,33], it was decided to investigate the Fenton activity of a range of fatty acids, namely stearic acid, oleic acid, and palmitic acid, which are all present in the human diet, together with the shorter-chained, myristic, lauric, and hexanoic acids.

Fatty acids possess low solubility in water, so it was decided to use a detergent to facilitate the dissolution of the various fatty acids under investigation. Initially, we selected sodium dodecyl sulfate (SDS; 2.5 mM), but we observed that the presence of SDS reduced the efficiency of Fe(III) EDTA to produce ^•^OH; furthermore, there were some light-scattering effects resulting from the SDS micelles. We reasoned that the sulphate groups on the surface of the micelles were chelating a proportion of the iron(III) in the reaction medium, so we investigated several non-charged detergents, namely triton-X100, Brig-58, and polyoxymethylene sorbitan monooleate. Again, with each detergent, we experienced variable degrees of light scattering in the 430 nm region of the spectrum. We then selected CTAB (cetyltrimethylammonium bromide) for investigation, reasoning that the positively charged detergent would not bind iron(III) and that the relatively low molecular weight would generate small micelles. Indeed, we discovered that CTAB (10 mM) did not interfere with the Fenton activity of Fe(III)EDTA, and so we adopted this detergent for the investigation into the Fenton activity of fatty acids in the presence of iron(III). 

It was clear from the results (Table 2) that the iron(III) complexes of fatty acids lack Fenton activity. Their activity was found to be similar to that of the oligodentate complexes of citric acid (Table 1 and Figure 2).

## 3. Discussion

As indicated in the Introduction section, there is a need for a relatively simple assay for the Fenton activity of metal complexes. Such a method was proposed in 1988 [22], but the assay has not been widely adopted. Recently, our group became interested in the potential application of iron(III) complexes for the treatment of anaemia and, as a consequence, reinvestigated the potential of NPBG as a probe for Fenton activity. We report here several modifications to the original synthesis of NPBG and more precise guidelines for the assay.

The application of UV/visible spectroscopy for monitoring Fenton activity is straightforward when using NPBG; the method can be used for a fixed-time assay, monitoring the solutions at 435 nm. Neither NPBG nor its hydroxylated products bind iron(III) at pH 8.0, so the probe does not compete for iron(III) present in the iron complexes under the conditions of the assay.

Regarding NPBG synthesis (Figure 1), we were unable to achieve the reported yields by using the previously published method [22]. However, by extending the reaction time and changing the workup for the isolation of the intermediate, 3,5-diamino-nitrobenzene, we achieved much improved yields. The subsequent conversion of 3,5-diaminonitrobenzene to NPBG was achieved as previously reported [22].

The precise procedure for the assay was not described in detail in the 1988 publication, and the sequence of additions to the assay solution proves to be critical. If the iron complex is formed in situ, this should be undertaken in unbuffered water: iron(III) chloride or iron(III) nitrate are dissolved in water (0.5 mL), and when the iron salt is completely dissolved, the chelator can be added. When the solution is complete, 9.5 mL of sodium borate buffer (25 mM, pH 8.0) is added, yielding a final iron(III) concentration of 0.5 mM. This solution is maintained at room temperature for 30 min and then briefly centrifuged. At this stage, H_2_O_2_ can be added in order to initiate the hydroxylation of NPBG. Using this procedure, we confirmed the previously reported results, using a range of chelators (Table 1). Overall, we improved the synthetic method for the production of NPBG and the analytical procedure for monitoring Fenton activity over and above that previously reported [22]. 

With fatty acids, the presence of detergents was necessary in order to completely dissolve the iron(III) complexes. CTAB was found to be applicable in the NPBG-based assay. Having confirmed that the presence of CTAB (10 mM) did not influence the Fenton activity of Fe(III)EDTA, we investigated a small range of fatty acids in the same system. To our surprise, no Fenton activity was detected with the iron(III) complexes of stearic acid (C_18_), myristic acid (C_14_), lauric acid (C_12_), and hexanoic acid (C_6_) (Table 2). Indeed, the fatty acids behaved like citric acid (Table 1). It has been demonstrated that citric acid forms stable oligo-iron(III) complexes [25], and it seems likely that fatty acids also form stable oligodentate complexes with iron(III). Indeed, such complexes have been reported for acetic and stearic acids [34,35]. The lack of Fenton activity of the iron(III) complexes of fatty acids renders them an attractive proposition for the oral treatment of anaemia.

## 4. Materials and Methods

### 4.1. Reagents and General Procedures

3,5-Dinitroaniline was purchased from Fluorochem. Sodium sulfide nonahydrate, glutaric anhydride, ethanol, ethyl acetate, hexane, acetonitrile, methanol, dimethyl sulfoxide-d6, hydrogen peroxide solution (30% *w*/*v*), ethylenediaminetetraacetic acid, ferric nitrate nonahydrate, fatty acids, diethylenetriaminepentaacetic acid, nitrilotriacetic acid, citric acid, maltol, and deferiprone were purchased from Sigma-Aldrich. Ammonium chloride and silica gel (0.006–0.200 mm, 60 Å) were purchased from Acros Organics. Sand for solid-phase extraction, sodium tetraborate decahydrate, and boric acid were purchased from Fisher Scientific. All detergents were purchased from Sigma-Aldrich. Reaction extracts were dried over anhydrous Na_2_SO_4_, and the solvents were removed under reduced pressure. Reactions were monitored by thin-layer chromatography (TLC), using commercial plates (Merck) precoated with silica gel 60 F-254. Visualisation was performed by UV fluorescence (λmax = 254 nm). Chromatographic separations on silica gel columns (Kieselgel 40, 0.040–0.063 mm; Merck) were performed by flash chromatography. Yields refer to chromatographically and spectroscopically pure compounds, unless otherwise stated. Compounds are named following IUPAC rules as applied by Beilstein-Institut AutoNom 2000 (4.01.305) or CA Index Name. All melting points were determined on a microscope hot-stage Büchi apparatus and are uncorrected. ^1^H-NMR and ^13^C-NMR spectra were obtained on a Bruker AVANCE 400 spectrometer at 400 MHz and 101 MHz, respectively, using an internal deuterium lock at ambient probe temperatures, in 5 mm i.d. glass tubes. Chemical shift (δ) values are quoted as parts per million (ppm) to the nearest 0.01 ppm (for ^1^H NMRs) or 0.1 ppm (for ^13^C NMRs), and they are referenced according to the residual non-deuterated solvent peak (i.e., DMSO-d6 at 2.50 ppm for proton and 39.5 ppm for carbon). The coupling constants (J) are reported in Hz. Data are reported as follows: chemical shift, multiplicity (br = broad, s = singlet, d = doublet, t = triplet, m = multiplet, q = quartet, quin = quintet, or as a combination of these, e.g., dd, dt, etc.), integration, assignment, and coupling constant(s). Assignments were determined either on the basis of an unambiguous chemical shift or coupling pattern, or by analogy to fully interpreted spectra for related compounds. UV–Vis spectra of the probes were recorded using a Perkin-Elmer Lambda 25 UV–Vis spectrophotometer. For HPLC analysis, a Hewlett-Packard Hypersyl RPC18 column (5 μm, 10 cm × 4.6 mm ID) was used to resolve NPBG and its *ortho-* and *para-*hydroxylated isomers. A linear gradient (0–10 of B) was used, where A = 20 mM phosphate buffer (pH = 7) and B = acetonitrile. Diode array detection allowed for monitoring over the range of 220 to 450 nm, with fixed channels at 232 and 430 nm. For MS analysis, stock solutions (1 mg/mL in MeOH) where diluted with 0.1% HCOOH in MeOH/H_2_O (50:50) to a final concentration of 50 µg/mL prior to being analysed. The instrument used consisted of a Thermo Accela LC system interfaced to a Thermo TSQ Access triple quadrupole mass spectrometer with a HESI source. The data were processed with Xcalibur™ Software (version 2.0). Then 10 µL of sample was analysed in flow injection: flow, 0.2 mL/min; and mobile phase, 0.1% HCOOC in MeOH/H2O (50:50). The parameters used for the analysis in the negative-ion mode were as follows: spray voltage, 3500 V; vaporizer temperature, 300 °C; sheath gas pressure, 50 au; and capillary temperature, 350 °C; capillary offset, -35. Microanalyses (C, H, and N) were performed by Medac Ltd. (Surrey, UK), and the results are within ±0.4% of the theoretical values.

### 4.2. Synthesis of N’N-(5-Nitro-1,3-phenylene)-bis-glutaramide (NPBG)

There were two steps to synthesise *N’N*-(5-nitro-1,3-phenylene)-*bis*-glutaramide (NPBG), i.e., the preparation of 3,5-diaminonitrobenzene and the subsequent amidation reaction between 3,5-diaminonitrobenzene and glutaric anhydride to produce NPBG (NMR and MS spectra are reported in the Appendix A).

*3,5-Diaminonitrobenzene.* 3,5-Dinitroaniline (460 mg) was dissolved in 5 mL of hot ethanol, forming a bright yellow solution. Ammonium chloride (500 mg) was dissolved into 1.5 mL of water. Sodium sulfide nonahydrate (1.54 g) was also dissolved into 1.5 mL of water. The ammonium chloride solution was slowly added to the dinitroaniline solution over the period of 5 min. The sodium sulfide solution was then added dropwise into the stirred mixture, which was maintained between 60 and 70 °C for a period of 30 min. In the process of adding sodium sulfide, the colour of the solution turned a brownish red with a yellow solid precipitating. After 30 min, the source of heating was removed. The pH of the mixture was adjusted to 8.0 with 2M hydrochloric acid, and the mixture was cooled to room temperature. The product was extracted with 15 mL of ethyl acetate, yielding an orange solution. The removal of the solvent by rotary evaporation under 40 °C and 240 mbar yielded an orange-red solid. This product was purified by flash column chromatography (hexane: ethyl acetate—1:3, silica gel). Rotary evaporation yielded red needle shaped crystals of 3,5-diaminonitrobenzene. The crystals were recrystallised from water and dried over P_2_O_5_ to yield 3,5-diaminonitrobenzene (70%). Mp 143–144 °C. ^1^H NMR (400 MHz, DMSO-d6) δ 6.59 (qd, 2H, ArH, J = 1.3 Hz, J = 0.8 Hz), 6.13 (dq, 1H, ArH, J = 2.7 Hz, J = 1.1 Hz), 5.40 (s, 4H, NH_2_). ^13^C NMR (101 MHz, DMSO-d6) δ 150.2 (1C, amine-ArC), 149.5 (1C, nitro-ArC), 104.1 (1C, ArC), 96.5 (2C, ArC).

*5,5′-[(5-nitro-1,3-phenylene)bis(azanediyl)]bis(5-oxopentanoic acid) (= N’N-(5-nitro-1,3-phenylene)-bis-glutaramide, NPBG).* Glutaric anhydride (45.3 mg) and 3,5-diaminonitrobenzene (285 mg) were dissolved in 15 mL of acetonitrile and heated under reflux for 1 h. The solvent was removed by rotary evaporation, giving a yellow-orange solid. The product was recrystallised from water, producing 490 mg (yield: 69%) of yellow fluffy powder. Mp 191–192 °C. ^1^H NMR (400 MHz, DMSO-d6) δ 12.08 (s, 2H, COOH), 10.38 (s, 2H, NH), 8.25 (d, 3H, ArH, J = 1.0 Hz), 2.39 (t, 4H, amide-CH_2_, J = 7.4 Hz), 2.28 (t, 4H, COOH-CH_2_, J = 7.3 Hz), 1.81 (quin, 4H, CH_2_*CH_2_*CH_2_, J = 7.4 Hz). ^13^C NMR (101 MHz, DMSO-d6) δ 174.1 (2C, COOH), 171.5 (2C, amide), 148.1 (1C, nitro-ArC), 140.4 (2C, amide-ArC), 114.6 (1C, ArC), 107.8 (2C, ArC), 35.4 (2C, amide-CH_2_), 32.9 (2C, COOH-CH_2_), 20.2 (1C, CH_2_). Elemental analysis calculated for C_16_H_19_N_3_O_8_ with 50.39% C, 5.02% H, and 11.01% N; found 50.05% C, 4.94% H, and 11.02% N. MS (ESI^−^) calculated for C_16_H_19_N_3_O_8_ 381.12, found 380.03 (*m*/*z*) [*M-H*]^−^, 402.01 (*m*/*z*) [*M+Na-2H*]^−^, and 266.02 [M(*-* CO(CH_2_)_3_COOH)*-*H]^−^. IR: v_max_ (neat)/cm^−1^ 3400 (O-H stretching), 3300 (N-H stretching), 1679 (C=O stretching), 1553 (NO_2_ asymmetric stretching), and 1335 (NO_2_ symmetric stretching). UV–Vis (water): λ_max_ at 230 nm (ε = 31,536 L·mol^−1^·cm^−1^), λ at 290 nm (ε = 3818 L·mol^−1^·cm^−1^).

### 4.3. Hydroxylation assay for N’N-(5-Nitro-1,3-phenylene)-bis-glutaramide (NPBG)

The solution adopted for the assay was buffered with sodium tetraborate (25 mM, pH 8.0) and contained NPBG (1 mM), iron complex (0.5 mM), and H_2_O_2_ (5 mM). H_2_O_2_ was excluded in the blank reference solution. The preparation of the iron complex and the sequence of additions are both critically important. The iron salt (either iron(III) or iron(II)) is dissolved in water (0.5 mL); when completely dissolved, the chelator can be added at the prescribed molar ratio to that of iron. When the chelator is completely dissolved, sodium borate buffer (25 mM, pH 8.0, 9.5 mL) containing NPBG (1.05 mM) is added, yielding a final iron concentration of 0.5 mM. The solution is maintained at room temperature for 30 min and then centrifuged (3000 rpm) for 5 min. The supernatant fluid is decanted. The hydroxylation is initiated by the addition of H_2_O_2_. The absorption at 435 nm is recorded after 1 h.

## Data Availability

Data is contained within the article or Appendix A, or available on request from the corresponding author.

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
