# Peer review of "The Colorimetric Detection of the Hydroxyl Radical"

_ijms, 2023, doi:10.3390/ijms24044162_

Round 1
Reviewer 1 Report
As the title implies, the authors set out improve the approach for the detection of hydroxyl radicals which is based on the UV-Vis absorption characteristic of a synthetic molecular probe. The work is interesting as detection and quantitation of hydroxyl radicals produced within a certain environment is indeed difficult to perform and the methods are seldom selective or specific, respectively. The manuscript reads well and the study is well designed. Authors present a new approach to prepare the N’N-(5-nitro-1,3-phenylene)-bis-glutaramide molecular probe that offers high selectivity and is therefore of potentially high academic (and industrial) value. The molecular probe has been described before and only critical modifications are reported in the current work. In this respect, the advancements made and gains achieved should be clearly highlighted in order to unambiguously present the novelty that would warrant the publication of the authors’ current work. The presented results are supported by the data, therefore, I recommend the work to be published after minor revision when the advancements to the field are clearly communicated.
Specific comments and questions:
1. Lines 41-42: even hydroxylation of aromatics is not always specific as was recently shown for coumarin and terephthalic acid, as both probe compounds reacted via more than one pathway (DOI: 10.1016/j.apcata.2020.117566). Authors should revise the text and include a short discussion on these issues which can significantly contribute to analytical bias.
2. Lines 55-62: other reports relevant to the study at hand should also be mentioned here (e.g., DOI: 10.1021/acs.est.2c03799, and others).
3. Line 74-75: Why does the ortho product not form? Due to the steric hindrances of the substituents or due to electronic effects? Perhaps both?
4. Figure 1 (left hand side): the numbers in the legend should be explained although I assume that they represent the incubation time.
5. Line 100: the absorbance increases with the incubation time, therefore, wouldn’t it be more reproducible to wait a certain amount of time for the absorbance to reach a plateau? Otherwise, small fluctuations in incubation time could lead to invalid analytical data.
6. Abbreviations should be explained in text when first used.
7. Line 214: Authors probably mean anhydrous Na2SO4?
8. Section 4.3: it is not clear at which point NPBG is added. Please revise.
9. Was the hydroxylated product(s) characterized? NMR? MS? Do only the monohydroxylated products form? Where is the data?
10. Section 4.3: the order of chemical additions is reversed according to the procedure described in 2.2. The authors should clarify this.
11. Section 2.2.: how was the concentration of the iron complex determined if some of the salt/chelator was removed by centrifugation? Does this not effect the measurements/kinetics?
12. Lines 116-127: I am not sure I understand. Shouldn’t a higher K1 values for FeII (strong complexes) translate into lower rates of hydroxylation when FeII is involved? Table 1 reports quite the opposite. And vice versa, I would expect high rates when FeIII forms strong complexes with chelators. Please elaborate.
13. Line 125: the authors probably mean “accessable”?
14. How were the iron(III) complexes of fatty acids determined/detected?
15. Lines 182-190: should this part be moved (in part) in section 4 (experimental)?
Author Response
Manuscript ID: ijms-2218684
We thank the referees for their feedback and comments on our manuscript. Our reply to each query follows.
Preamble
In this respect, the advancements made and gains achieved should be clearly highlighted in order to unambiguously present the novelty that would warrant the publication of the authors’ current work. The presented results are supported by the data, therefore, I recommend the work to be published after minor revision when the advancements to the field are clearly communicated.
We have rephrased the abstract and added a sentence to increase clarity in this regard. We have also added further explanation in this regard in paragraph 3, Discussion.
Specific comments and questions:
- Lines 41-42: even hydroxylation of aromatics is not always specific as was recently shown for coumarin and terephthalic acid, as both probe compounds reacted via more than one pathway (DOI: 10.1016/j.apcata.2020.117566). Authors should revise the text and include a short discussion on these issues which can significantly contribute to analytical bias.
We have modified extensively the text (see lines 46-74) to reflect the reviewer suggestion and cited the above reference, along with other 2 relevant references in this context.
- Lines 55-62: other reports relevant to the study at hand should also be mentioned here (e.g., DOI: 10.1021/acs.est.2c03799, and others).
See above, Re: 2 references cited.
- Line 74-75: Why does the ortho product not form? Due to the steric hindrances of the substituents or due to electronic effects? Perhaps both?
We believe the text does not need to be changed. Here we refer to the catechol function that can form e.g. in salicylate like structures. We have expressely mentioned this in the revised manuscript. In NPBG two hydroxy group in ortho to each other are not possible. We indeed obtain both ortho- and para-hydroxylated derivatives (see answer to next comments) but neither are part of “ortho di-hydroxy” function. This is important as an “ortho di-hydroxy” function would coordinate metal ions, including iron(III) and iron(II), both in general terms and under the conditions of the assay.
- Figure 1 (left hand side): the numbers in the legend should be explained although I assume that they represent the incubation time.
This has been amended. For clarity, incubation time is now mentioned in the legend.
- Line 100: the absorbance increases with the incubation time, therefore, wouldn’t it be more reproducible to wait a certain amount of time for the absorbance to reach a plateau? Otherwise, small fluctuations in incubation time could lead to invalid analytical data.
We trust we have established accurate conditions for a “fixed time assay”, which is generally considered to be the most reliable and convenient form of assay for this type of work. We believe it is certainly not advisable to wait for a plateau to be reached as this would most likely introduce variations.
- Abbreviations should be explained in text when first used.
This has been amended. We have checked the manuscript and expressed abbreviations at first appearance where missing.
- Line 214: Authors probably mean anhydrous Na2SO4?
This has been amended.
- Section 4.3: it is not clear at which point NPBG is added. Please revise.
We have added this experimental detail in section 4.3.
- Was the hydroxylated product(s) characterized? NMR? MS? Do only the monohydroxylated products form? Where is the data?
We have added a section in the main text (lines 120-127) for the characterisations by HPLC and UV for the 2 mono-hydroxylated products that form from NPBG and provided the spectral data into supplementary information.
- Section 4.3: the order of chemical additions is reversed according to the procedure described in 2.2. The authors should clarify this.
This has been clarified, please see paragraph 2.2 lines 107-112.
- Section 2.2.: how was the concentration of the iron complex determined if some of the salt/chelator was removed by centrifugation? Does this not effect the measurements/kinetics?
Under the defined conditions, the iron will not precipitate and so the iron concentration is known. We found centrifugation advisable to remove only dust particulate which may scatter light.
- Lines 116-127: I am not sure I understand. Shouldn’t a higher K1 values for FeII (strong complexes) translate into lower rates of hydroxylation when FeII is involved? Table 1 reports quite the opposite. And vice versa, I would expect high rates when FeIII forms strong complexes with chelators. Please elaborate.
Yes, in principle high K1 values will produce lower hydroxylation rates, but with EDTA and NTA (as explained in the text) the 1:1 complexes both have a vacant coordination site for H2O2 to attack. As both bind Fe(II) and Fe(III), they can redox cycle. We have only minimally changed the text.
- Line 125: the authors probably mean “accessible”?
We have changed the word to accessible.
- How were the iron(III) complexes of fatty acids determined/detected?
The iron(III) complexes of fatty acids form well defined oligomeric complexes in stoichiometric amounts. We give reference to this property in Refs. 34, 35. Thus, the concentration of the iron complexes with fatty acids was determined by the amount of iron added to the solution in the same manner as it is achieved with for instance EDTA or Deferiprone.
- Lines 182-190: should this part be moved (in part) in section 4 (experimental)?
We have adopted the guidelines/template for IJMS and included in experimental only technicalities/protocols, we would prefer to leave this more narrative part in the current section.
Reviewer 2 Report
In this work, a spectrophotometric method based on the hydroxylation of the substrate N, N'- (5-nitro-1,3-phenylene)bisglutaramide was developed for the determination of hydroxyl radicals. In general, this manuscript was well organized, highlighting the research significance of this work. However, there are still some issues that need to be properly revised before publication. My suggestions are as follows:
1 Please keep the format uniform for hydroxyl radical. Except for the first use of hydroxyl radical (•OH) in the line 27, Such as line 67, hydroxyl radical should be replaced by •OH.
2. The synthesized probe NPBG should be characterized by FTIR to further confirm the successful synthesis.
3. line 165, As indicated in the Introduction, here, Introduction should be revised to “Introduction”.
4. Section 4. Materials and Methods should be moved to before 2. Results may be better.
5. The reaction condition should be provided in the figure 1.
Author Response
Manuscript ID: ijms-2218684
We thank the referees for their feedback and comments on our manuscript. Our reply to each query follows.
1 Please keep the format uniform for hydroxyl radical. Except for the first use of hydroxyl radical (•OH) in the line 27, Such as line 67, hydroxyl radical should be replaced by •OH.
This has been amended.
- The synthesized probe NPBG should be characterized by FTIR to further confirm the successful synthesis.
We have added the FTIR spectrum to SI.
- line 165, As indicated in the Introduction, here, Introduction should be revised to “Introduction”.
This has been amended throughout.
- Section 4. Materials and Methods should be moved to before 2. Results may be better.
We have used the IJMS template and followed the guidelines, which suggest adopting current formatting for submissions to this journal.
- The reaction condition should be provided in the figure 1.
Full reaction conditions and reagents are provided in the legend of Figure 1.